# Isoquinolinequinone Derivatives from a Marine Sponge (*Haliclona* sp.) Regulate Inflammation in In Vitro System of Intestine

**DOI:** 10.3390/md19020090

**Published:** 2021-02-04

**Authors:** Yun Na Kim, Yeong Kwang Ji, Na-Hyun Kim, Nguyen Van Tu, Jung-Rae Rho, Eun Ju Jeong

**Affiliations:** 1Department of Agronomy and Medicinal Plant Resources, Gyeongnam National University of Science and Technology, Jinju 52725, Korea; yunna@gntech.ac.kr; 2Department of Oceanography, Kunsan National University, Gunsan 54150, Korea; kwang7089@kunsan.ac.kr; 3Gyeongnam Department of Environment & Toxicology, Korea Institute of Toxicology, 17 Jegok-gil, Munsan-eup, Jinju-si 52834, Korea; nhkim@kitox.re.kr; 4Institute of Tropical Biology, 85 Tran Quoc Toan Street District 3, Ho Chi Minh 700000, Vietnam; nguyenvan.tu@itb.ac.vn

**Keywords:** isoquinolinequinone, *Haliclona* sp., inflammation, intestine, inflammatory bowel disease, co-culture

## Abstract

Using bio-guided fractionation and based on the inhibitory activities of nitric oxide (NO) and prostaglandin E2 (PGE2), eight isoquinolinequinone derivatives (**1**–**8**) were isolated from the marine sponge *Haliclona* sp. Among these, methyl *O*-demethylrenierate (**1**) is a noble ester, whereas compounds **2** and **3** are new *O*-demethyl derivatives of known isoquinolinequinones. Compound **8** was assigned as a new 21-dehydroxyrenieramycin F. Anti-inflammatory activities of the isolated compounds were tested in a co-culture system of human epithelial Caco-2 and THP-1 macrophages. The isolated derivatives showed variable activities. *O*-demethyl renierone (**5**) showed the highest activity, while **3** and **7** showed moderate activities. These bioactive isoquinolinequinones inhibited lipopolysaccharide and interferon gamma-induced production of NO and PGE2. Expression of inducible nitric oxide synthase, cyclooxygenase-2, and the phosphorylation of MAPKs were down-regulated in response to the inhibition of NF-κB nuclear translocation. In addition, nuclear translocation was markedly promoted with a subsequent increase in the expression of HO-1. Structure-activity relationship studies showed that the hydroxyl group in **3** and **5**, and the N-formyl group in **7** may be key functional groups responsible for their anti-inflammatory activities. These findings suggest the potential use of *Haliclona* sp. and its metabolites as pharmaceuticals treating inflammation-related diseases including inflammatory bowel disease.

## 1. Introduction

Marine sponges are considered as prolific sources of novel compounds exhibiting diverse biological activities, which can be used for the treatment of human diseases [1,2,3]. Isoquinolinequinones, a class of metabolites derived from marine sponges, belonging to the genera *Xesotospongia* and *Riniera*, and to *Petrosia* sp., have been shown to possess a variety of biological activities, including cytotoxic, antimicrobial, antifungal, and antineoplastic effects [4,5,6]. During screening for in-house compounds isolated from various marine sponges with potent anti-inflammatory activities, some isoquinolinequinones isolated from the methanolic extract of *Haliclona* sp. showed significant nitric oxide (NO) and prostaglandin E2 (PGE2) inhibitory activities in THP-1 macrophages activated by lipopolysaccharide (LPS) and interferon-gamma (IFNγ). In analysis of the ^1^H NMR spectrum of the methanolic extract of *Haliclona* sp., it was expected that various isoquinolinequinones derivatives were contained. In the previous study, anti-inflammation of the extract of *Haliclona* sp. was reported in LPS-activated Raw 264.7 macrophages [7]. The extract of *Haliclona* sp. inhibited the production of NO and interleukin-1β induced by LPS. However, secondary metabolites contained in *Haliclona* sp. that contribute to anti-inflammatory activity have not been identified. In the present study, we focused on the identification of novel isoquinolinequinones from *Haliclona* sp., and the evaluation of their anti-inflammatory activities using an in vitro co-culture system resembling an intestine.

Recently, inflammatory bowel disease (IBD) has emerged as a threat to public health worldwide [8]. IBD, including Crohn’s disease (CD) and ulcerative colitis (UC), are chronic idiopathic disorders causing inflammation of the gastrointestinal tract in a genetically susceptible host [9]. CD is characterized by the appearance of non-continuous inflammatory lesions that can be found in any part of the gastrointestinal tract, from the mouth to the anus. In contrast, UC is characterized by a limited inflammatory region that usually begins in the rectum and spreads in a continuous manner. IBD is associated with genetic, environmental, and immune factors [10]. The exact cause of IBD is not fully known, but its pathology is well understood [11]. The structure and functions of intestinal tissue are maintained by numerous cells. These cells communicate with and are influenced by each other. Intestinal epithelial cells are firmly connected to form a semi-permeable barrier, but weakening of this epithelial barrier increases the permeability of the intestinal mucosa to pathogens, leading to the initiation of IBD [12]. The development of IBD is believed to be associated with disturbances in intestinal homeostasis, such as aberrant immune responses and subsequent exaggerated production of inflammatory mediators [13].

The in vitro intestinal model (combining epithelial cells and macrophages) mimics the complexity of the mucosal immune system [14]. Caco-2, human epithelial cells, are known to have transport and permeability characteristics similar to human intestinal tissue [15,16]. In this study, the anti-inflammatory activities of isoquinolinequinones were evaluated using an in vitro co-culture system consisting of epithelial Caco-2 cells and differentiated THP-1 macrophages. In addition, key functional groups responsible for intestinal anti-inflammatory effects were deduced based on structure-activity relationship (SAR) studies of the isoquinolinequinone derivatives.

## 2. Results and Discussion

Using bioactivity-guided fractionation, eight isoquinolinequinones (**1**–**8**), including four new derivatives, were isolated (Figure 1). Eight compounds, with a 5,8-dioxo-5,8-dihydroisoquinoline moiety, were isolated from a sponge *Haliclona* sp. The structures of the following compounds were identified using NMR and HR-MS experiments: methyl *O*-demethylrenierate (**1**, a new compound), *O*-demethylrenierol (**2**, a new derivative), 1,6-dimethyl-7-hydroxy-5,8-dihydroisoquinoline-5,8-dione (**3**, a new derivative), 1,6-dimethyl-7-methoxy-5,8-dihydroisoquinoline-5,8-dione (**4**) [17], *O*-demethylrenierone (**5**) [17], renierone (**6**) [6], *N*-formyl-1,2-dihydrorenierone (**7**) [17], and 21-dehydroxyrenieramycin F (**8,** a new derivative) (Appendix A). 

### 2.1. Structures of Compounds ***1***–***3*** and ***8*** Isolated from Haliclona sp.

The molecular formula of methyl *O*-demethylrenierate (**1**), a pale reddish amorphous powder in methanol, was deduced as C_12_H_9_NO_5_ based on the molecular ion peak at *m*/*z* 248.0554 [M + H]^+^ in the positive HRqTOF MS and the ^13^C NMR spectra. This molecular formula corresponds to nine degrees of unsaturation. The IR spectrum showed absorption peaks at 3400 and 1732 cm^−1^, indicating the presence of hydroxy and carbonyl groups. The UV spectrum exhibited absorption bands at 253, 299, and 325 nm. The ^1^H NMR spectrum showed the only four signals (two doublets and two singlets), whereas the ^13^C NMR spectrum showed 12 signals, including 1 methyl, 1 methoxy, 2 methines, and 8 non-protonated carbon signals through interpretation of the edited HSQC spectrum (Table 1). The downfield-shifted non-protonated carbons and the degrees of unsaturation suggested that **1** has a planar structure with aromaticity. Based on the HMBC correlations of the methyl proton at δ_H_ 2.15 and the two doublet protons at δ_H_ 8.06 and 9.04 with non-protonated carbons as shown in Figure 2a, an unassigned carbon chemical shift (δ_C_ 179.2), and the carbon chemical shifts of C-1 and C-3 suggested an isoquinolinequinone unit with a hydroxy group at C-7. The carbon and proton chemical shifts for the unit were similar to those of the known renierol, which is an isoquinolinequinone [18]. Additionally, the remaining carbonyl carbon at δ_H_ 166.6 was correlated with the methoxy methyl protons at δ_H_ 4.08. This carbon did not show any correlation with the isoquinolinequinone, but the connection with C-1 could complete the entire structure of **1**. Thus, compound **1** was deduced as methyl 7-hydroxy-6-methyl-5,8-dioxo-5,8-dihydroisoquinoline-1-carbonate. The structure of **1** was confirmed based on the similarity of the carbon chemical shifts calculated using the Density functional theory (DFT) method with MPW1PW91/6-311 + G(2d,p)//B3LYP/6-31+G(d,p), after conformational searching using Merck molecular force field (MMFF). Figure 2b shows the slight differences in the calculated and measured carbon chemical shifts. 

The molecular formula of *O*-demethylrenierol (**2**) was determined as C_11_H_9_NO_4_ based on the positive HRqTOF-MS and the ^13^C NMR spectra. Its NMR spectral data were found to be similar to those of the known compound, renierol. Compared to renierol, **2** has a molecular weight difference of 14 [18]. The methoxy signal in the ^1^H NMR spectrum of **2** disappeared, suggesting the replacement of the methoxy group by a hydroxy group at the position of C-7. Similarly, the HR-MS data and the ^1^H NMR spectrum showed that, compared to compound **4** (1-dehydroxyrenierol), compound **3** was substituted by a hydroxy group at C-7. Compound **3** was named 1,6-dimethyl-7-hydroxy-5,8-dihydroisoquinoline-5,8-dione.

The molecular formula of 21-dehydroxyreniermycin F (**8**) was determined to be C_31_H_36_N_2_O_9_ based on the positive HRqTOF-MS and the ^13^C NMR spectra (Appendix A). Interpretation of the 1D and 2D NMR spectra revealed that the structure of **8** was similar to that of renieramycin F, with the absence of a hydroxy group at position C-21 [19]. 

### 2.2. Anti-Inflammatory Effects of Compounds ***1***–***8*** in THP-1 Macrophages Activated Using Lipopolysacchride (LPS) and IFNγ

The anti-inflammatory activities of the isolated compounds (**1–8**) were evaluated in THP-1 macrophages activated using LPS (10 μg/mL) and IFNγ (10 ng/mL). The inhibitory effects of the compounds on the LPS + IFNγ- induced production of NO and PGE2 were measured. Prior to assays, cytotoxicity of compounds **1**–**8** against THP-1 macrophages and Caco-2 cells were tested. Although compound **8** showed cytotoxicity at a concentration over 1 μM in CCK-8 assays, the cell viability of the cells treated with compounds **1**–**7** at a concentration below 10 μM was found to be similar to that of the non-treated controls (>95% in control group, Appendix A). Following the cytotoxicity test, THP-1 cells were treated with compounds **1**–**7** (1, 5, and 10 μM) or **8** (0.1, 0.5, and 1 μM) for 1 h, and then, LPS + IFNγ was added. After 24 h of incubation with LPS + IFNγ, the levels of NO and PGE2 released into the culture medium were measured (Figure 3 and Figure 4). The levels of NO and PGE2 in the media of cells treated with compounds **1**–**8** at the highest concentrations (10 μM for **1**–**7** and 1 μM for **8**) are summarized in Table 2. LPS + IFNγ treated cells, which were pretreated with compounds **3**, **5**, **6**, and **7** for 24 h, showed a significant decrease in NO production of 59.4, 49.0, 57.7, and 52.6% respectively. In addition, the decrease in PGE2 production was most significant in cells treated with compound **5**.

### 2.3. Effects of Compounds ***1***–***8*** on Pro-Inflammatory Protein Expression and MAPK Phosphorylation in THP-1 Macrophages Co-Cultured with Caco-2 Cells

A significant increase in the expression of pro-inflammatory proteins, such as iNOS and COX-2, is observed in the pathogenesis of colitis [20]. Overexpressed iNOS and COX-2 triggers the production of NO and PGE2 in activated macrophages, leading to inflammatory processes [21,22]. To evaluate the efficacy of compounds **1**–**8** in regulating inflammation in an in vitro intestinal co-culture system, the expression of iNOS and COX-2 was evaluated using Western blotting. Figure 5 shows that the expression levels of iNOS and COX-2 were significantly upregulated by treatment with LPS + INFγ. In cells pretreated with compounds **3**, **5**, and **7** (1, 5, 10 μM), the induced expressions of iNOS and COX-2 induced by LPS + INFγ were significantly attenuated in a concentration-dependent manner. In cells pretreated with compound **2**, only the expression of iNOS was significantly attenuated with no significant changes observed in the expression of COX-2.

Subsequently, we evaluated the inhibitory effects of compounds **1–8** on the phosphorylation of proteins of the mitogen-activated protein kinase (MAPK) family, including p38, ERK1/2, and JNK (Figure 6). The increased phosphorylation of p38, ERK1/2, and JNK in the LPS + IFNγ-treated cells was inhibited by pretreatment with compounds **3**, **5** and **7**. In addition, compound **2** also inhibited the phosphorylation of MAPKs at the highest concentration (10 μM). The most significant reduction in p38, ERK1/2, and JNK phosphorylation was observed in cells treated with compound **5**.

### 2.4. Effects of Compounds ***1***–***8*** on the Nuclear Translocation of NF-κB and Its Inhibitor, IκB-α

The transcription factor, NF-κB, is generally known to play a key role in immune and inflammatory signaling by controlling the expression of related genes [23,24]. Because the binding sites for NF-κB are located proximally to the COX-2 and TNF-α promoter genes, the inhibition of the DNA-binding activity of NF-κB induces a suppression of the expression of the inflammatory mediators, iNOS, COX-2, and TNF-α in macrophages [25,26,27]. Based on previous results showing the inhibitory effects of **3**, **5** and **7** on the expression of pro-inflammatory cytokines and proteins, the regulatory effects of these compounds on the localization of p65, a specific gene set of NF-κB, and phosphorylated p65 in the cell nucleus and cytoplasm were evaluated. As shown in Figure 7, Western blotting showed that **2**, **3**, **5**, **6** and **7** (1, 5, 10 μM) inhibited the phosphorylation of p65 in the cytoplasm and its translocation into nuclei. In addition, pretreatment with these compounds significantly upregulated the total levels of the IκB-α protein in the cytoplasm. This observation may account for the enhanced binding of IκB-α to p65, which prevents the entry of NF-κB into the nucleus to bind to the promoter of the gene. The order of compound bioactivities with respect to the translocation of p65 and cytoplasmic expression of IκB-α was as follows: **5** >> **3**, **7**, **2**, **6** > **4**, **8**, **1**. Our results provide evidence that these active compounds disrupt the interactions of the p65 subunit of NF-κB with specific sets of target genes, which consequently downregulates the subsequent transcription and expression of pro-inflammatory mediators [28,29].

### 2.5. Effects of Compounds ***3***, ***5***, and ***7*** on the Expression of HO-1 in THP-1 Macrophages Co-Cultured with Caco-2 Cells

HO-1 is known to suppress the expression of pro-inflammatory proteins and cytokines in activated macrophages [30,31]. Recently, the importance of Nrf-2/HO-1 signaling as an anti-inflammatory effector was demonstrated in animal models of IBD. In mouse models of colitis induced by dextran sulfate sodium (DSS), marked enhancement in the expression of Nrf-2/HO-1 was observed during colitis-wound repair [32]. In a 2,4,6-trinitrobenzenesulfonic acid-induced mouse model of colitis, the administration of lyophilized biomass of the microalga, *Chlamydomonas debaryana,* was shown to increase the expression of Nrf-2 and HO-1 [33]. To evaluate the involvement of **3**, **5**, and **7** in the regulation of Nrf-2/HO-1 signaling, Western blotting was conducted. As shown in Figure 8, treatment with all three compounds at the tested concentrations (1, 5, and 10 μM) increased the levels of HO-1 in co-cultured Caco-2 cells and THP-1 macrophages. Nuclear factor [erythroid-derived 2]-like 2 (Nrf-2) is a critical regulator of the coordinated induction of phase II enzymes, including HO-1, and achieves this effect via binding to the antioxidant response element. Following exposure to these compounds, the nuclear levels of Nrf2 sharply increased, with a concomitant decrease in cytoplasmic levels. In this study, we observed that **3**, **5**, and **7** markedly promoted the nuclear translocation of Nrf2 and the subsequent increase in the expression of HO-1.

### 2.6. Structure-Activity Relationship of the Haliclona sp. Isoquinolinequinone Derivatives 

Based on the anti-inflammatory activity results of the 8 compounds using an in vitro co-culture system, a structure-activity relationship (SAR) study was conducted. Among the isoquinolinequinone derivatives (**1**–**8**), compound **5** showed significantly superior activities. Compounds **2**, **3**, and **7** showed moderate activities, whereas **1**, **4**, **6**, and **8** either showed weak activities or were inactive. The order of bioactivity of the compounds was as follows: **5** >> **3**, **7** > **2** > **6**, **4**, **8**, **1.** Interestingly, compounds **3**, **4**, **5**, and **6** had similar chemical structures with the exception of the substituents at C-7 (-OH or -OCH_3_). Concerning the expression of iNOS, COX-2, and MAPKs, and NF-κB signaling, significant decrease in activity was observed for compounds **4** and **6** as compared to compounds **3** and **5**. Compounds **3** and **5**, which possess a hydroxy group at C-7, were active, whereas **4** and **6**, which possess a methoxy group at C-7, showed either weak activity or were inactive, indicating that the presence of the hydroxy group at C-7 may play an important role in regulation of the inflammatory factor. Moreover, although compound **7** possessed a methoxy group at C-7, it showed potent anti-inflammatory activities in all assays. Comparison of compounds **6** and **7** strongly suggested that the N-formyl group in **7** may be a key functional group responsible for activity. Compound **7** (*N*-formyl-1,2-dihydrorenierone) has been recently reported in the marine sponge *Xestospongia* sp. [34]. Our SAR analysis showed that compound **7**, compared to other derivatives, also has potent NK-κB inhibitory activity without cytotoxicity. In line with the previous report, it was concluded that the N-formyl group is important for the regulation of inflammation [34]. Since the number of compounds that were tested was limited, the SAR analysis could only suggest that the hydroxyl group at C-7 and the N-formyl group at N-2 could play a key role in regulating inflammatory factors in macrophages.

## 3. Materials and Methods

### 3.1. General Procedures

Measurement of optical rotation and IR spectra were performed using a JASCO P-1010 polarimeter and FT/IR 4100 spectrometer (JASCO Corporation, Tokyo, Japan), respectively. UV spectra were recorded in MeOH using a Varian Cary 50 Bio spectrometer (Varian Inc., Palo Alto, CA, USA). HRqTOFMS were carried out using a SCIEX X500R (Sciex Co., Framingham, MA, USA). The NMR spectra were recorded using a Varian VNMRS 500 spectrometer at 500 and 125 MHz for ^1^H and ^13^C, respectively (Varian Inc., Palo Alto, CA, USA). Compounds were isolated and purified using high-performance liquid chromatography (HPLC) with an YMC ODS-A column (250 × 10 mm, 5 μm) and a phenomenex C8 column (250 × 10 mm, 5 μm). DFT calculation was made on a Dell PowerEdge R740 Server (Dell, Rounf Rocks, TX, USA) with gaussian 16 (Gaussian. Inc., Wallingford, CT, USA) and spartan’18 softwares (Wavefunction Inc., Irvine, CA, USA) installed. 

### 3.2. Materials

LPS (*Escherichia coli* 0127: B8), dimethyl sulfoxide (DMSO), and PMA (2-mercaptoethanol, phorbol 12-myristate 13-acetate) were purchased from Sigma-Aldrich (St. Louis, MO, USA). MEM and RPMI 1640 media, fetal bovine serum (FBS), penicillin/streptomycin and phosphate-buffered saline (PBS) were purchased from Gibco Life Technologies (Grand Island, NY, USA). Griess reagent was purchased from (Promega, Madison, WI, USA). Enzyme-linked immunosorbent assay (ELISA) kits for PGE2 were purchased from R&D Systems (Minneapolis, MN, USA). Primary antibodies against iNOS, COX-2, p65, Lamin B, phospho-p65, IκBα, phospho-ERK, ERK, phospho-p38, p38, phospho-JNK, JNK, HO-1 and Nrf2 were purchased from Cell Signaling Technology, Inc. (Beverly, MA, USA). Primary anti-β-actin monoclonal antibodies and secondary antibodies were purchased from Sigma-Aldrich (St. Louis, MO, USA).

### 3.3. Animal Material

The sponge *Haliclona* sp. (phylum Porifera, class Demospongiae, order Haplosclerida, family Chalinidae) was collected in April 2018, Vietnam (09°56′09.0″ N, 104°01′07.4″ E). A voucher specimen (MABIK Lot No. 0014027) was deposited at the Marine Biodiversity Institute of Korea (MABIK) and was authenticated by Dr. Young-A Kim at Hannam University. Sponge samples were immediately frozen and kept at −25 °C until sample extraction.

### 3.4. Isolation of Isoquinolinequinones from Haliclona sp.

The freeze-dried specimen (2.5 kg) was extracted twice using MeOH (2 L) at 25 °C for 24 h, and partitioned between H_2_O and CH_2_Cl_2_ for desalting. The CH_2_Cl_2_ and H_2_O layers were repartitioned into 85% aqueous MeOH and hexane, and H_2_O and butanol, respectively. After combing the 85% aqueous MeOH and butanol components, the solvent was evaporated and the residue (1.8 g) was subjected to reverse phase open column chromatography using stepwise elution, starting from 50% H_2_O (I) to 100% MeOH (VI), with gradual increments of 10% MeOH, to elute six fractions. Three fractions (III, 172 mg; IV, 155 mg; and V, 199 mg) that showed NO inhibitory activity were separated using HPLC to yield eight compounds. A mixed fraction was isolated from fraction III using HPLC (YMC ODS-A, 250 × 10 mm; 50% H_2_O + 50% MeOH; RI detector; 2 mL/min) and was again separated to give red-colored compounds, namely, **1** (2.0 mg) and **2** (1.8 mg) using HPLC (Phenomenex C8, 250 × 10 mm; 65% H_2_O + 35% ACN; RI detector; 2 mL/min). Compound **3** (2.1 mg) was isolated from fraction IV using HPLC (YMC ODS-A 250 × 10 mm; 30% H_2_O + 70% MeOH; RI detector; 2 mL/min). Compounds **4** (17 mg), **5** (5.3 mg), **6** (7.1 mg), **7** (5.5 mg), and **8** (5.1 mg) were isolated from fraction V using HPLC (Phenomenex C8, 250 × 10 mm; 30% H_2_O + 70% MeOH; RI detector; 2 mL/min). The isolated compounds were purified using HPLC (>95%). 

Methyl *O*-demethylrenierate (**1**): a reddish-brown powder; UV (MeOH) λ_max_ (log ε) 253 (3.60), 299 (3.33), 325 (2.93) nm; IR ν_max_ (film) 3400, 2914, 1732, 1458 cm^−1^; ^1^H and ^13^C NMR see Table 1; HRqTOFMS *m*/*z* 248.0554 [M + H]^+^ (cald for C_12_H_10_NO_5_, 248.0553). 

*O*-demethylrenierol (**2**): a reddish-brown powder; UV (MeOH) λ_max_ (log ε) 252 (3.62), 294 (3.16), 317 (3.06) nm; IR ν_max_ (film) 3310, 2921, 1734, 1461 cm^−1^; ^1^H and ^13^C NMR see Table 1; HRqTOFMS *m*/*z* 220.0584 [M + H]^+^ (cald for C_11_H_10_NO_4_, 220.0604).

1,6-dimethyl-7-hydroxy-5,8-dihydroisoquinoline-5,8-dione (**3**): a reddish-brown powder; UV (MeOH) λ_max_ (log ε) 250 (3.48), 295 (3.11), 320 (2.92) nm; IR ν_max_ (film) 3400, 2921, 1734, 1457 cm^−1^; ^1^H and ^13^C NMR see Table 1; HRqTOFMS m/z 204.0652 [M + H]^+^ (cald for C_11_H_10_NO_3_, 204.0655).

21-dehydroxyrenieramycin F (**4**): a red powder; ${\left[\alpha \right]}_{D}^{25}$ = + 32.0 (*c* 0.1, MeOH); UV (MeOH) λ_max_ (log ε) 208 (4.21), 269 (4.09) nm; IR ν_max_ (film) 3330, 2928, 1719, 1656, 1232 cm^−1^; ^1^H and ^13^C NMR see Appendix A; HRqTOFMS m/z 581.2494 [M + H]^+^ (cald for C_31_H_37_N_2_O_9_, 581.2494).

### 3.5. Quantum Calculations of ^13^C NMR Chemical Shifts

The carbon chemical shifts of **1** were calculated by conformational searching under Merck molecular force field (MMFF) and the DFT method using the Spartan’16 and the Gaussian 16 software respectively (Appendix A) [35]. Conformers with relative energy below 5 kcal/mol were searched and optimized using DFT calculation at the B3LYP level with 6-31G(d) basis set. One conformer, with a prevalent Boltzmann distribution, was chosen, and the ^13^C NMR calculation was performed at the MPW1PW91/6-311G(d) level using the polarizable continuum model in chloroform. 

### 3.6. Cell Cultures

The human epithelial cell line (Caco-2) was purchased from the American Type Culture Collection (ATCC, Manassas, VA, USA). The cells were cultured in MEM (Gibco BRL, Grand Island, NY, USA) containing 20% (*v/v*) fetal bovine serum (FBS) and 1% (*v/v*) antibiotics (100 U/mL penicillin and 100 μg/mL streptomycin). The human monocytic cell line (THP-1) was purchased from the American Type Culture Collection (ATCC, Manassas, VA, USA). The cells were cultured in RPMI1640 (Gibco BRL, Grand Island, NY, USA), containing 10% (*v/v*) FBS, 1% (*v/v*) antibiotics (100 U/mL penicillin and 100 μg/mL streptomycin) and 0.05 mM 2-mercaptoethanol. The cells were incubated in a humidified atmosphere of 95%-air and 5%-CO_2_.

### 3.7. Differentiation of THP-1 to Macrophages

Differentiation of THP-1 cells with PMA are widely used as a model for function of human macrophages [12]. Induction of differentiation in THP-1 cells was carried out using 50 ng/mL PMA for three days, followed by incubation in a fresh medium without PMA for two more days.

### 3.8. In Vitro Intestinal Co-Culture Model

To establish the human intestinal co-culture system, Caco-2 cells were seeded at 3.75 × 10^5^ cells/well on transwell inserts (pore size: 0.4 μm; Corning CoStar Corp., Cambridge, MA, USA) and maintained for 14–20 days in an incubator at 37 °C in a 5% CO_2_ atmosphere. The culture medium was changed every 3 days until the cells were fully differentiated: transepithelial electrical resistance (TEER) value > 1200 Ω cm^2^. THP-1 cells were independently seeded at 8.5 × 10^6^ cells/well onto the plate bottom of a 6-well transwell plate. Then, the polarized Caco-2 monolayer insert was added. The upper and lower chambers would represent the apical and basolateral sides of the intestinal epithelium, respectively [36,37,38] (Appendix A). To evaluate anti-inflammatory activity, LPS (10 µg/mL) and interferon-gamma (IFNγ, 10 ng/mL) were added to the basolateral compartment of the plate. The isolated compounds were dissolved in dimethyl sulfoxide at the concentration of 100 mg/mL, and diluted in culture medium before use. The compounds to be tested were added to the apical compartment, at the indicated concentrations. After incubation for an appropriate period of time, the levels of the produced inflammatory mediators in the culture supernatant from the basolateral side were measured.

### 3.9. TEER Measurement

Measurement of TEER values was performed, using a Millicell-ERS (Millipore, MA, USA), to assess monolayer integrity. The TEER values were calculated as following: TEER(Ω cm^2^) = Resistance–Blank resistance (Ω) × Membrane surface Area (cm^2^). During the development of the co-culture models, TEER values are extensively used to measure the resistance of the tight junctions of cell monolayers [36]. In our co-culture system, the TEER of the Caco-2 layer significantly increased 4 days after plating and reached the maximum value at day 21.

### 3.10. Cell Viability Assay

Cell viability was measured using the CCK-8, Cell Counting Kit-8 (Donjinjo Molecular Biology, Inc., Kumamoto, Japan), according to the manufacturer’s instructions. Briefly, THP-1 cells were seeded at a density of 1 × 10^4^ cells/well in 96-well plates and incubated for 24 h. Then, the cells were treated with each compound at various concentrations for 24 h. After incubation, 10 μL of CCK-8 solution was added to each well and incubated for 3 h at 37 °C. The absorbance was measured using a microplate reader (Bio-Tek Company, Winooski, VT, USA) at 450 nm. The experiments were performed in triplicate.

### 3.11. Measurement of NO and PGE2 Production

THP-1 cells were seeded at 1 × 10^4^ cells/well in 96-well plates and incubated for 24 h. The cells were treated with the test sample for 1 h, followed by treatment with LPS (10 µg/mL) and interferon-gamma (IFNγ, 10 ng/mL) for 24 h. The amount of nitrite produced in the culture medium was quantified using a Griess assay. Briefly, the 100 μL of culture supernatants was mixed with 50 μL of sulfanilamide solution and 50 μL of N-1-naphthylethylenediamine dihydrochloride solution. This was then incubated for 10 min at room temperature, protected from light. After 15 min of incubation, the optical density was determined at 540 nm with a microplate reader (Bio-Tek Company, Winooski, VT, USA). The amount of NO was quantified from calibration curve of sodium nitrite. The concentration of prostaglandin E2 (PGE2) was measured using an enzyme-linked immunosorbent assay (ELISA) kit (R&D System, Minneapolis, MN, USA) according to the manufacturer’s instructions. The culture medium used for the ELISA assay was not diluted.

### 3.12. Western Blot

The co-culture system was established as indicated in Section 3.6 (Appendix A). After incubation with test compounds and/or LPS + IFNγ, THP-1 cells from the basolateral compartment were washed three times with ice-PBS and extracted with a radioimmunoprecipitation (RIPA) buffer containing a protease inhibitor cocktail (Santa Cruz, CA, USA) for 40 min on ice. Protein extracts were centrifuged at 13,000× *g* for 30 min at 4 °C. Thirty micrograms of the lysed proteins were separated using SDS-PAGE (8–12%) at 100 V and transferred to polyvinylidene fluoride (PVDF) membranes. The membranes were blocked with 5% non-fat milk in a phosphate buffered saline with tween 20 (PBST) buffer for 1 h at room temperature. These were then incubated with the respective primary antibodies at 4 °C overnight. The incubated membranes were washed three times with a PBST buffer, and incubated further with secondary antibodies for 1 h at room temperature. Bands were visualized using ECL solution (Thermo Fisher Scientific, Waltham, MA, USA) and calibrated using the Chemidoc Imaging System (Bio-Rad, Hercules, CA, USA).

### 3.13. Statistical Analysis

Data were analyzed using Prism version 5.00 (GraphPad Software, San Diego, CA, USA). Student’s *t*-test and Dunnett’s post-hoc test was performed following a significant one-way ANOVA to determine the level of significance of the parameters measured in the different groups. *p* < 0.05 was considered as statistically significant. Data of three independent experiments are expressed as mean ± SD.

## 4. Conclusions

Eight isoquinolinequinones (**1**–**8**) were isolated from the marine sponge *Haliclona* sp. from Vietnam. Among them, four compounds were new derivatives. Compound **1** was new and was characterized through oxidation based on the presence of a carbonyl group at C-11. Compounds **2** and **3** were identified as derivatives formed by the replacement of the methoxy group with a methyl group at C-7 in renierol and 1,6-dimethyl-7-methoxy-5,8-dihydroisoquinoline-5,8-dione, respectively. Compound **8** was identified as a compound formed by the removal of a hydroxy group at C-21 of renieramycin F. The anti-inflammatory effects of all compounds were evaluated in a co-culture system established by combining epithelial Caco-2 cells and THP-1 macrophages, which mimicked the intestinal environment. Among these compounds, **5** exhibited the most significant anti-inflammatory activity. Compounds **2, 3**, and **7** showed moderate activities, whereas compounds **1**, **4**, **6**, and **8** showed either weak activity or were inactive. These bioactive isoquinolinequinone derivatives regulated the expression of iNOS and COX-2, and the MAPK family (ERK, p38, JNK). Nuclear translocation of NF-κB, and the expression of Nrf-2 and HO-1 were also regulated. SAR analysis showed that, although compound **1** showed weak activity, the hydroxy group at C-7 and the N-formyl group at N-2 may be key functional groups responsible for anti-inflammatory activity. Considering that NF-κB could be regarded as the master switch in the regulation of inflammation in macrophages, this study supports the potential use of *O*-demethylisoquinolinequinones as a therapeutic agents in the treatment of IBD. 

## Figures and Tables

**Figure 1 marinedrugs-19-00090-f001:**
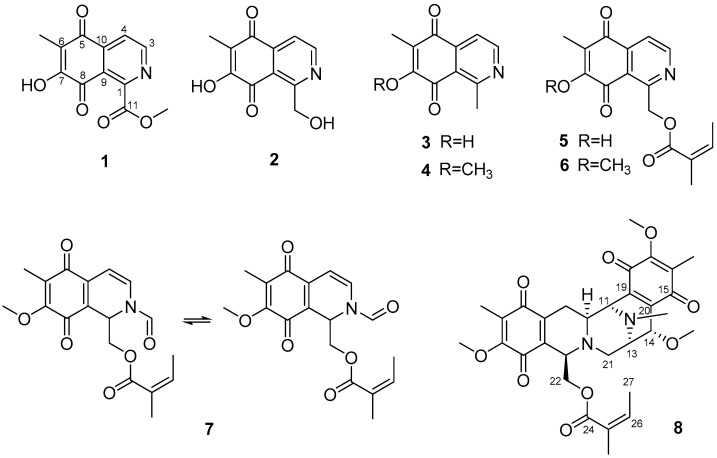
Structures of the isoquinolinequinones (**1**–**8**) isolated from *Haliclona* sp.

**Figure 2 marinedrugs-19-00090-f002:**
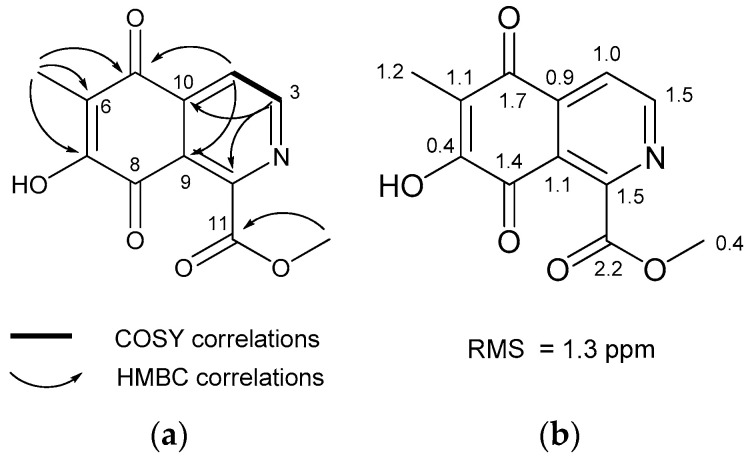
(**a**) Key COSY and HMBC correlations; (**b**) Difference between the calculated and experimental ^13^C NMR chemical shifts in compound **1.**

**Figure 3 marinedrugs-19-00090-f003:**
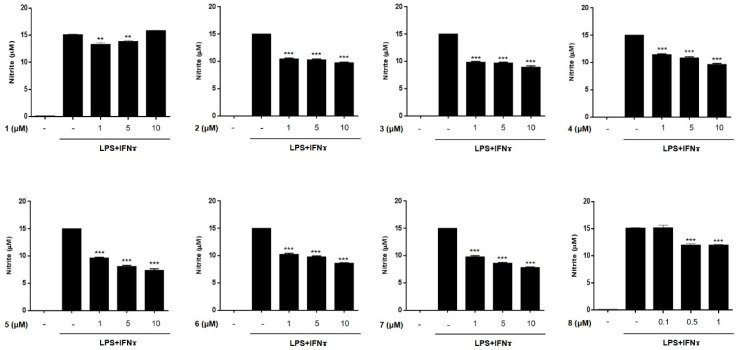
Effects of compounds **1**–**8** on the production of NO in THP-1 cells activated by LPS + IFNγ. THP-1 macrophages were treated with each compound (**1**–**8**) at three concentrations (1, 5, 10 μM for compounds **1**–**7**; 0.1, 0.5, 1 μM for compound **8**), followed by LPS (10 μg/mL) + IFNγ (10 ng/mL) were treated. After 24 h of incubation, the concentrations of NO released into culture medium were measured using a Griess assay as described in materials and methods. Results are presented as the means ± SDs of triplicate experiments; ** *p* < 0.01 and *** *p* < 0.001 compared to LPS + IFNγ-treated cells. LPS: lipopolysaccharide; IFNγ:interferon-gamma.

**Figure 4 marinedrugs-19-00090-f004:**
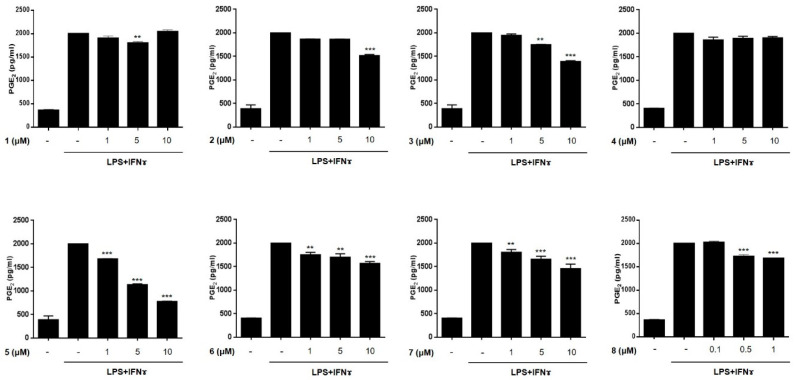
The effects of compounds **1**–**8** on the production of PGE2 in THP-1 cells activated by LPS + IFNγ. THP-1 macrophages were treated with each compound (**1**–**8**) at three concentrations (1, 5, 10 μM for **1**–**7**; 0.1, 0.5, 1 μM for **8**), followed by LPS (10 μg/mL) + IFNγ (10 ng/mL) were treated. After 24 h of incubation, the content of PGE2 released into culture medium was measured using ELISA kit. Results are presented as the means ± SDs of triplicate experiments; ** *p* < 0.01 and *** *p* < 0.001 compared to LPS + IFNγ-treated cells. LPS: lipopolysaccharide; IFNγ: interferon-gamma.

**Figure 5 marinedrugs-19-00090-f005:**
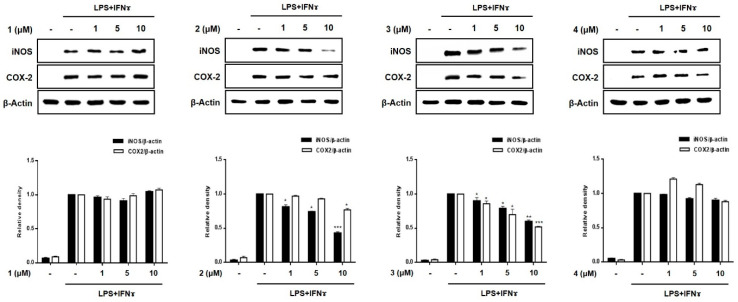
Effects of compounds **1**–**8** on the expression of the pro-inflammatory proteins, iNOS and COX-2 in a co-culture system of Caco-2 cells and THP-1 macrophages. Compounds **1**–**8** were individually added into the apical compartment, while LPS (10 μg/mL) + IFNγ (10 ng/mL) was added to the basolateral compartment of the Caco-2/THP-1 co-culture model. After 24 h of incubation, the expression of iNOS and COX-2 in the pretreated cells was analyzed using Western blotting. The representative (upper panel) and quantified blots (bottom panel) were obtained after normalization to β-actin. Results are presented as the means ± SDs of triplicate experiments; * *p* < 0.05, ** *p* < 0.01 and *** *p* < 0.001 compared to LPS + IFNγ -treated cells. LPS: lipopolysaccharide; IFNγ: interferon-gamma.

**Figure 6 marinedrugs-19-00090-f006:**
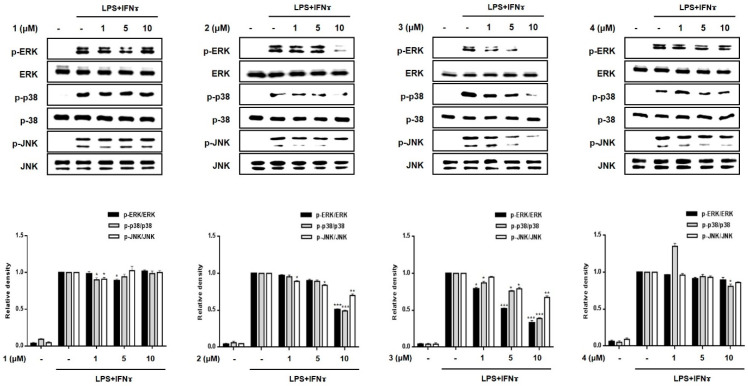
Effects of compounds **1**–**8** on the phosphorylation of MAPKs in a co-culture system of Caco-2 cells and THP-1 macrophage. Compounds **1**–**8** were individually added into the apical compartment, while LPS (10 μg/mL) + IFNγ (10 ng/mL) was added to the basolateral compartment of the Caco-2/THP-1 co-culture model. After 24 h of incubation, the phosphorylation levels of MAPKs (ERK, p38, JNK) in cells treated with the compounds were measured using Western blotting. The level of phospho-form of ERK, p38, and JNK was normalized to the total level of each protein in quantified blots (bottom panel). Results are presented as the means ± SDs of triplicate experiments; * *p* < 0.05, ** *p* < 0.01 and *** *p* < 0.001 compared to LPS + IFNγ-treated cells. LPS: lipopolysaccharide; IFNγ: interferon-gamma.

**Figure 7 marinedrugs-19-00090-f007:**
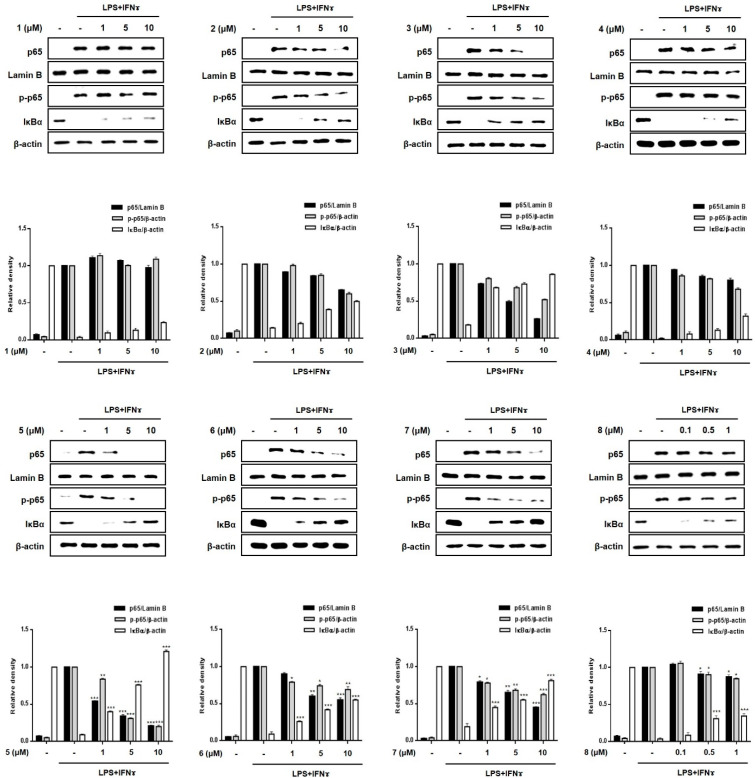
Effects of compounds **1**–**8** on the nuclear translocation of NF-κB in a co-culture system of Caco-2 cells and THP-1 macrophages. Compounds **1**-**8** were individually added into the apical compartment, and LPS (10 μg/mL) + IFNγ (10 ng/mL) was added to the basolateral compartment of the Caco-2/THP-1 co-culture model and incubated overnight. Cell extracts were biochemically separated into nuclear and cytoplasmic fractions, and the expression of p65, phosphorylated-p65, Lamin B and IκBα in cells treated with each compound were analyzed using Western blotting. The representative (upper panel) and quantified blots (bottom panel) were obtained after normalization to Lamin B (nucleus) or β-actin (cytosol). Results are presented as the means ± SDs of triplicate experiments; * *p* < 0.05, ** *p* < 0.01 and *** *p* < 0.001 compared to LPS + IFNγ-treated cells. LPS: lipopolysaccharide; IFNγ: interferon-gamma.

**Figure 8 marinedrugs-19-00090-f008:**
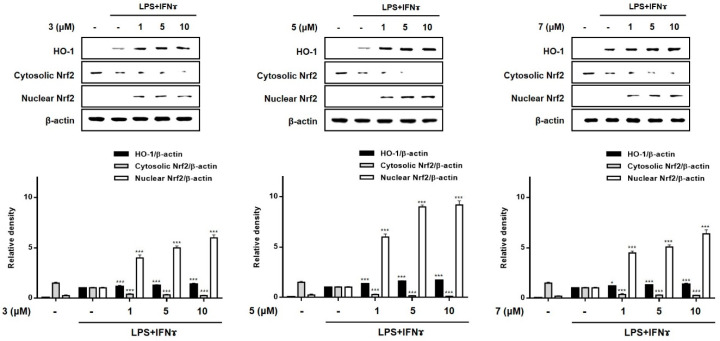
Effects of compounds **3**, **5**, and **7** on the expression of HO-1 and nuclear translocation of Nrf2 in a co-culture system of Caco-2 cells and THP-1 macrophages. Each compound (1, 5, and 10 μM) was individually added into the apical compartment, and LPS (10 μg/mL) + IFNγ (10 ng/mL) was added to the basolateral compartment of the Caco-2/THP-1 co-culture model. After incubating for 12 and 1 h, respectively, the expression of HO-1 and Nrf2 in cells was analyzed using Western blotting. The representative (upper panel) and quantified blots (bottom panel) were obtained. The representative blot is shown (left panel) and calculated intensities are shown (right panel) after normalization to Lamin B (nucleus) or β-actin (total protein or cytosol). Results are presented as the means ± SDs of triplicate experiments; * *p* < 0.05 and *** *p* < 0.001 compared to LPS + IFNγ-treated cells. LPS: lipopolysaccharide; IFNγ: interferon-gamma.

**Table 1 marinedrugs-19-00090-t001:** Spectral data for compounds **1**–**3** in CDCl_3_ (500MHz for ^1^H, 125 MHz for ^13^C).

No.	1		2		3	
	^13^C, mult	^1^H, m(*J* Hz)	^13^C, mult	^1^H, m(*J* Hz)	^13^C, mult	^1^H, m(*J* Hz)
1	150.6, C		160.4, C		160.5, C	
3	155.6, CH	9.04, d(4.9)	154.1, CH	9.00, d(4.9)	155.1, CH	8.91, d(4.9)
4	120.5, CH	8.06, d(4.9)	118.9, CH	7.99, d(4.9)	118.0, CH	7.88, d(4.9)
5	182.8, C		183.6, C		184.0, C	
6	121.9, C		120.4, C		121.0, C	
7	153.4, C		153.5, C		153.7, C	
8	179.2, C		181.1, C		181.6, C	
9	120.1, C		119.7, C		119.7, C	
10	139.0, C		140.1, C		140.0, C	
11	166.6, C		64.0, CH_2_	5.22, d(4.2)	25.6, CH_3_	3.02, s
6-CH_3_	8.8, CH3	2.15, s	8.6, CH_3_	2.12, s	8.5, CH_3_	2.11, s
OCH_3_	53.4, CH3	4.08, s				

**Table 2 marinedrugs-19-00090-t002:** Inhibitory effects of compounds **1**–**8** (10 μM) on LPS + IFN-γ-induced production of NO or PGE2 in THP-1 macrophages.

Compounds (10 μM)	NO	PGE2
	Relative Content (%) vs. LPS + IFN-γ-Treated Cells	
**1**	100.0	102.1
**2**	63.3	76.5
**3**	59.4	70.6
**4**	64.2	95.4
**5**	49.0	39.1
**6**	57.7	78.3
**7**	52.6	73.6
**8** *	80.2	84.2

* The concentration of compound **8** was set to 1 μM owing to its significant cytotoxicity at a concentration of 10 μM in CCK-8 assay.

## Data Availability

Not Applicable.

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
