# Peer review of "Isoquinolinequinone Derivatives from a Marine Sponge (Haliclona sp.) Regulate Inflammation in In Vitro System of Intestine"

_marinedrugs, 2021, doi:10.3390/md19020090_

Round 1

Reviewer 1 Report

The manuscript evaluates the anti-inflammatory activity of several isoquinolinequinone derivatives from marine sponge Haliclona sp. in a co-culture system of Caco-2 and THP-1 cells. In particular, the manuscript show a systematic approach to evaluate the ability of the isoquinolinequinone derivatives to modulate different cellular and molecular parameters of inflammation elicited by lipopolysaccharide and interferon gamma in THP-1 cells. The results seem interesting but their presentation is neglected and incomplete.

  • They should describe the preparation of the compound stock and the solvent used in materials and methods.
  • They report the Griess Reagent System in materials but not use it. The Griess Reagent System measures nitrite but the authors use DAF-DA assay. What does DAF-DA mean? The authors should review these sections of materials and methods.
  • How do they calculate the amount of nitrite in figure 3? They should report these informations in materials and methods.
  • They should correct wrong informations in the legends of Figure 4 and 8.
  • The authors should report cytotoxicity data of the isoquinolinequinone derivatives as supplemental material.
  • The authors report inhibitory effects of the isoquinolinequinone derivatives in Table 2. In this regard, It is not unclear the data expression. They could express these data in terms of the percentage of inhibition of the NO and PGE2 release.
  • They describe the statistical analysis in materials and methods but not perform it. Therefore, they should perform the statistical analysis and report the significance levels in results and discussion section as well as in figures 3-8. In this regard, they should also describe the number of experiments performed and the post hoc test used with one-way ANOVA in the legends.

Author Response

Dear reviewer,

We thank you for providing the helpful and insightful comments to improve our manuscript. Our point-by-point response to your comments is provided below. The modified parts are marked in Yellow. Also, we notice that the manuscript has been carefully edited for spelling, grammar, and readability using Editage’s language editing service.

  1. They should describe the preparation of the compound stock and the solvent used in materials and methods.

→ The preparation of the compound was described in materials and methods (section 3.8.)

  1. They report the Griess Reagent System in materials but not use it. The Griess Reagent System measures nitrite but the authors use DAF-DA assay. What does DAF-DA mean? The authors should review these sections of materials and methods.

→ The assay used to measure nitrite concentration was DAF-FM diacetate assay, not Griess assay. The typos in materials were corrected. In addition, the brief method of assay was described in section 3.11.

  1. How do they calculate the amount of nitrite in figure 3? They should report these informations in materials and methods.

→ The amount of nitrite was quantified by measuring DAF-FM fluorescence with fluorescence microplate reader. The figure legend 3 was modified.

  1. They should correct wrong informations in the legends of Figure 4 and 8.

→ Thank you for the comments. The legends of figure 4 and 8 were revised.

  1. The authors should report cytotoxicity data of the isoquinolinequinone derivatives as supplemental material.

→ Cytotoxicity data of compounds 1-8 were given as supplemental material (S20). In addition, we additionally measured cytotoxicity of compounds against Caco-2 epithelial cells (S21).

  1. The authors report inhibitory effects of the isoquinolinequinone derivatives in Table 7. In this regard, It is not unclear the data expression. They could express these data in terms of the percentage of inhibition of the NO and PGE2 release.

→ As the Reviewer suggested, the term in Table 2 is modified.

  1. They describe the statistical analysis in materials and methods but not perform it. Therefore, they should perform the statistical analysis and report the significance levels in results and discussion section as well as in figures 3-8. In this regard, they should also describe the number of experiments performed and the post hoc test used with one-way ANOVA in the legends.

→ Statistical analysis was included in Figure 3~8. The number of experiments are described, and all figure legends are revised.

Reviewer 2 Report

The manuscript entitled “Isoquinolinequinone derivatives from a marine sponge (Haliclona sp.) regulate intestinal inflammation” describes structures and anti-inflammatory properties of 8 compounds isolated from a sponge Haliclona sp. This manuscript is the next in a series of studies published by these Authors, describing the anti-inflammatory activity of compounds isolated from marine sponges in the intestine inflammation model. Structures of isolated compounds were identified using NMR and HR-MS experiments. Anti-inflammatory properties of these compounds were studied in THP-1 macrophages co-cultured with Caco-2 cells and treated with LPS and IFN-γ. The Authors identified 4 new isoquinolinequinone derivatives with rather low or moderate biological activity compared to the most effective compound, O-demethylrenierone. These compounds decreased the LPS/IFN-induced production of NO and PGE2 and downregulated expression of COX-2 and iNOS. Authors postulated that hydroxyl group at C-7 and N-formyl group at N-2 could play a key role in determining the anti-inflammatory activity of isoquinolinequinone and O-demethylisoquinolinequinones should be taken into consideration as therapeutic agents in the treatment of IBD. The manuscript is interesting however, it has many drawbacks and in its present form does not meet the criteria of Marine Drugs. Moreover, in terms of methodology and result content, it is similar to two previous works published in Marine Drugs 2019 (doi:10.3390/md17110634) and Molecules 2019 (doi:10.3390/molecules24183394), and the Authors did not go further to reveal mechanisms of anti-inflammatory activity of isolated compounds.

  1. The title of the manuscript should be changed since the Authors studied the compound activity in the in vitro model of intestinal inflammation.
  2. There is no statistical analysis of the results even though the Methods includes a description of used tests (3.14). Appropriate statistic tests should be included and Figures must be completed.
  3. I recommend performing additional experiments concerning the activity of isolated compounds, especially these new ones. Apart from THP-1 also CaCo-2 viability should be checked to exclude a cytotoxic activity of compounds toward the epithelial cells. I suggest including the results of the cytotoxicity test in the manuscript. Moreover, the impact of LPS/IFN and isolated compounds on the IEC integrity/permeability should be examined since it is directly connected with intestine inflammation.
  4. Results are combined with Discussion in one chapter. They should be described separately, which will allow in-depth discussion of the results including a proposition of the mechanism of anti-inflammatory properties of the compounds.
  5. Titles 2.3 and 2.5 are inappropriate since these chapters describe the anti-inflammatory properties of compounds in THP-1 macrophages only not in both cell lines.
  6. The section Materials and Methods contains fragments which are inadequate to the Results content, e.g., ELISA test of cytokines, Griess reagent system, list of primers. In contrast, the method of fractionation of cell lysates into nuclear and cytoplasmic fractions is not described. Taking the above into account, the whole manuscript has to be carefully examined to evaluate such misleading fragments.
  7. The purity of isolated compounds should be estimated. Moreover, the solvent in which compounds were dissolved and applied to the cell media should be indicated and the concentration of stocks should be included in the Methods.
  8. If I am correct, in Figure 6 normalization was not performed to actin, as the figure legend stated, but to the total level of appropriate protein, e.g., p-ERK to ERK. Similarly, in Figure 7 normalization was performed to actin for proteins in cytoplasmic fraction and lamin for proteins in nuclear fraction.
  9. A scheme of the co-culture model and its stimulation would make the text easier to understand.
  10. The Authors used a high concentration of LPS 10 ug/ml plus interferon-gamma to induce inflammation of THP-1 cells. Such a high concentration of LPS is more appropriate for the induction of inflammatory response of CaCo-2 cells. The justification of usage of this LPS concentration should be included.
  11. Line 297 “PMA (2-mercaptoethanol, phorbol 12-myristate 12-acetate)” odd construction; PMA is an abbreviation of phorbol 12-myristate 13-acetate.
  12. The manuscript should be checked by English native speaker.

Minor

  • The term “alkaloid” is not mentioned anywhere in the text.
  • Line 335, 338, and other - probably “powder” instead of “power”.
  • Line 136 “insult with” ?
  • Reference to THP-1cells differentiation method should be added
  • Figure 7 “cell lysates” rather than “cell extracts”.

Author Response

Dear reviewer,

We thank you for providing the helpful and insightful comments to improve our manuscript. Our point-by-point response to your comments is provided below. The modified parts are marked in Yellow. Also, we notice that the manuscript has been carefully edited for spelling, grammar, and readability using Editage’s language editing service.

  1. The title of the manuscript should be changed since the Authors studied the compound activity in the in vitro model of intestinal inflammation.

→ As the reviewer suggested, the title of the manuscript is changed to  “Isoquinolinequinone derivatives from a marine sponge (Haliclona sp.) regulate inflammation in in vitro system of intestine”

  1. There is no statistical analysis of the results even though the Methods includes a description of used tests (3.14). Appropriate statistic tests should be included and Figures must be completed.

→ Statistical analysis was included in Figure 3~8. The number of experiments are described, and all figure legends are revised.

  1. I recommend performing additional experiments concerning the activity of isolated compounds, especially these new ones. Apart from THP-1 also CaCo-2 viability should be checked to exclude a cytotoxic activity of compounds toward the epithelial cells. I suggest including the results of the cytotoxicity test in the manuscript. Moreover, the impact of LPS/IFN and isolated compounds on the IEC integrity/permeability should be examined since it is directly connected with intestine inflammation.

→ The marine sponge, Haliclona sp. can only be obtained through collection from nature. The remained amount of compounds isolated from Haliclona sp. were not enough to conduct all assays the Reviewer recommended to perform additionally. As the Reviewer suggested, we further checked cytotoxicity of compounds (1~8) against Caco-2 epithelial cells. Similarly to results in THP-1 cells, compounds 1~7 showed no cytotoxicity at the concentration below 10 uM, whereas compound 8 showed the significant cytotoxicity at the concentration over 1 uM. Cytotoxicity data of compounds on THP-1 and Caco-2 cells are presented as supplementary data (S20 and S21).

  1. Results are combined with Discussion in one chapter. They should be described separately, which will allow in-depth discussion of the results including a proposition of the mechanism of anti-inflammatory properties of the compounds.

→ Thank you for your comments. The reason Results and Discussion sections are combined was that the previous reports on anti-inflammatory activities of Haliclona sp. or isoquinoline compounds are very limited. This paper is the first report to identify the secondary metabolites, isoquinoline derivaties that contribute to anti-inflammatory activity of Halinclona sp. The present study secured a variety of derivatives from Halicona sponges, and mainly focused on clarifying the structure-activity relationship of these derivatives. Instead of separately describing the Discussion, the results of SAR analysis were discussed in depth in section 2.6.

  1. Titles 2.3 and 2.5 are inappropriate since these chapters describe the anti-inflammatory properties of compounds in THP-1 macrophages only not in both cell lines.

→ As the Reviewer suggested, the title of 2.3 and 2.5 are modified. “…in co-culture system of THP-1 and Caco-2 cells” was changed to “…in THP-1 macrophages co-cultured with Caco-2 cells”

  1. The section Materials and Methods contains fragments which are inadequate to the Results content, e.g., ELISA test of cytokines, Griess reagent system, list of primers. In contrast, the method of fractionation of cell lysates into nuclear and cytoplasmic fractions is not described. Taking the above into account, the whole manuscript has to be carefully examined to evaluate such misleading fragments.

→ Thank you for comments. The section Materials and Methods is carefully revised. ELISA kit for IL-6 and TNF-alpha were deleted. Griess reagent was deleted and replaced with assay using DAF-FM diacetate. The list of primer is deleted.

  1. The purity of isolated compounds should be estimated. Moreover, the solvent in which compounds were dissolved and applied to the cell media should be indicated and the concentration of stocks should be included in the Methods.

→ The preparation of the compound stock was described in section 3.8. The isolated compounds 1~8 were purified using HPLC (>95%) which is described in section 3.4.

  1. If I am correct, in Figure 6 normalization was not performed to actin, as the figure legend stated, but to the total level of appropriate protein, e.g., p-ERK to ERK. Similarly, in Figure 7 normalization was performed to actin for proteins in cytoplasmic fraction and lamin for proteins in nuclear fraction.

→ In western blotting assay, the total level of proteins and their phosphorylated forms were each normalized to actin. In Figure 6, the quantified blot (bottom panel) was shown as ratio of phospho form/Total form of each protein. As the reviewer commented, in Figure 7 and 8, the nucleus fraction was normalized to lamin whereas total protein and cytosol fraction was normalized to actin. The legends of Figure 6~8 were revised.

  1. A scheme of the co-culture model and its stimulation would make the text easier to understand.

→ As the Reviewer suggested, a scheme of co-culture system is additionally given as supplementary data, S23.

  1. The Authors used a high concentration of LPS 10 ug/ml plus interferon-gamma to induce inflammation of THP-1 cells. Such a high concentration of LPS is more appropriate for the induction of inflammatory response of CaCo-2 cells. The justification of usage of this LPS concentration should be included.

→ Interferron-gamma and LPS are generally used stimuli to activate macrophages. Muller et al., (Frontiers in Immunology, 2016) reported that the treatment of IFN-r was shown to synergize with TLR agonist, LPS for induction of macrophage to produce NO and pro-inflammatory cytokines (TNF-a, IL-12). In our co-culture system, the concentration of IFN-r and LPS was optimized to activate THP-1 macrophages based on the production of NO and cytokines, and the expression of inflammatory mediators.

  1. Line 297 “PMA (2-mercaptoethanol, phorbol 12-myristate 12-acetate)” odd construction; PMA is an abbreviation of phorbol 12-myristate 13-acetate.

→ Thank you for your comments. The typos were corrected.

  1. The manuscript should be checked by English native speaker.

→ The manuscript has been carefully edited for spelling, grammar, and readability using Editage’s language editing service.

Minor points

The term “alkaloid” is not mentioned anywhere in the text.

→ “alkaloid” is deleted in keywords

Line 335, 338, and other - probably “powder” instead of “power”.

→ “power” was corrected to “powder”

Line 136 “insult with” ?

→ “insult with” was changed to “incubated with”

Reference to THP-1cells differentiation method should be added

→ A Reference about THP-1 differentiation using PMA is added

Figure 7 “cell lysates” rather than “cell extracts”.

→ “cell lysates” was changed to “cell extracts”

Round 2

Reviewer 1 Report

The authors have made the required changes. However, they should add some information in the method section:

  1. They expressed the amount of nitrite as μM (figure 3). Did they extrapolate these data from a standard curve of sodium nitrite? If yes, they should add these informations in method section.
  2. Did they use Bonferroni or Dunnett test for post hoc analysis of one-way ANOVA? They should add these informations in method section.

Author Response

Dear reviewer,

We thank you for providing the helpful and insightful comments to improve our manuscript. Our point-by-point response to your comments is provided below. The modified parts are marked in Blue.

  1. They expressed the amount of nitrite as μM (figure 3). Did they extrapolate these data from a standard curve of sodium nitrite? If yes, they should add these informations in method section.

→ Sorry for causing confusion. To optimize the experimental condition for NO measurement in THP-1 cells, both Griess and DAF-DA reagents are applied, and a reagent used for NO quantification in this study was Griess, not DAF-DA. The amount of nitrite was calculated from a calibration curve of sodium nitrite. The method was corrected.

  1. Did they use Bonferroni or Dunnett test for post hoc analysis of one-way ANOVA? They should add these informations in method section.

→ Thank you for your careful review. Dunnett test was used for post hoc analysis of one-way ANOVA. This information was added in method.

Reviewer 2 Report

The authors performed some additional experiments and included most of suggested changes in the manuscript, however, some of their corrections brought new inaccuracies. For example, the Authors added statistical analysis, however, in figure legends results were described as +/- SD whereas in point 3.13 of Materials and Methods results as +/- SEM. Results of cytotoxicity tests of studied compounds were included but without a description of CCK-8 assay and statistics. Moreover, descriptions of scales on the cytotoxicity graphs do not correspond to the values on the scales (Fig. S22). Authors corrected the spelling of the PMA name in one place, leaving the incorrect one in another place. In Fig. 6, if the level of proteins was normalized to actin, actin blots should be included in this figure.

Taking above into consideration I suggest Authors once more carefully check the manuscript.

Author Response

Dear reviewer,

We thank you for providing the helpful and insightful comments to improve our manuscript. Our point-by-point response to your comments is provided below. The modified parts are marked in Blue.

The authors performed some additional experiments and included most of suggested changes in the manuscript, however, some of their corrections brought new inaccuracies.

  1. The Authors added statistical analysis, however, in figure legends results were described as +/- SD whereas in point 3.13 of Materials and Methods results as +/- SEM.

→ Thank you for your comments. SEM in Materials and Methods was corrected to SD.

  1. Results of cytotoxicity tests of studied compounds were included but without a description of CCK-8 assay and statistics. Moreover, descriptions of scales on the cytotoxicity graphs do not correspond to the values on the scales (Fig. S22).

→ The method of CCK assay was described in Methods. Also, the description of scales on cytotoxicity graph is revised.

  1. Authors corrected the spelling of the PMA name in one place, leaving the incorrect one in another place.

→ The spelling of PMA name is revised and corrected throughout the manuscript.

  1. In Fig. 6, if the level of proteins was normalized to actin, actin blots should be included in this figure.

→ Thank you for your careful review. Actin blots are included in Figure 6.